# Research Progress of Safety of Zearalenone: A Review

**DOI:** 10.3390/toxins14060386

**Published:** 2022-06-02

**Authors:** Xiao Han, Bingxin Huangfu, Tongxiao Xu, Wentao Xu, Charles Asakiya, Kunlun Huang, Xiaoyun He

**Affiliations:** 1Key Laboratory of Precision Nutrition and Food Quality, College of Food Science and Nutritional Engineering, China Agricultural University, Beijing 100083, China; hanxiao2809@163.com (X.H.); bhuangfu@163.com (B.H.); tongxiao_xu23@163.com (T.X.); asakiya@cau.edu.cn (C.A.); 2Key Laboratory of Safety Assessment of Genetically Modified Organism (Food Safety), Ministry of Agriculture and Rural Affairs of the China, Beijing 100083, China; 06031@cau.edu.cn; 3Department of Nutrition and Health, China Agricultural University, Beijing 100083, China

**Keywords:** mycotoxin, zearalenone, risk assessment, toxicity, intestinal health, endocrine interference

## Abstract

Zearalenone, a mycotoxin produced by fungi of the genus *Fusarium*, widely exists in animal feed and human food. The structure of zearalenone is similar to estrogen, so it mainly has estrogenic effects on various organisms. Products contaminated with zearalenone can pose risks to animals and humans. Therefore, it is imperative to carry out toxicological research on zearalenone and evaluate its risk to human health. This paper briefly introduces the production, physical, and chemical properties of zearalenone and the research progress of its toxicity kinetics, focusing on its genetic toxicity, reproductive toxicity, hepatotoxicity, immunotoxicity, carcinogenicity, endocrine interference, and its impact on intestinal health. Finally, the progress of the risk assessment of human exposure is summarized to provide a reference for the follow-up study of zearalenone.

## 1. Introduction

Mycotoxins produced by fungi contaminate crops during their growth, maturation, harvesting, and storage. Zearalenone (ZEA), also known as F-2 toxin, is a common mycotoxin. It is the toxic metabolites produced by fungi of the genus *Fusarium*, such as *Fusarium graminearum*, *Fusarium oxysporum*, *Fusarium equisetum*, and *Fusarium nivalis* [1,2].

ZEA (molecular formula: C_18_H_22_O_5_) was first obtained from *Fusarium*-contaminated corn [3]. The melting point of ZEA is as high as 161–163 °C, with a weak polarity [4]. Normally, ZEA appears white crystalline. ZEA is fat-soluble and is almost insoluble in water and carbon tetrachloride solution. Its solubility gradually increases in n-hexane, benzene, acetonitrile, dichloromethane, methanol, ethanol, and acetone [5]. Additionally, ZEA is soluble in an alkaline aqueous solution. When the toxin is intaken by mammals, the ketone group at the C-8′ position of ZEA may reduce and transform to α- and β-zearalenol (zearalenol, ZOL), both of which have estrogen hormone-like structures (Figure 1) [6]. These substances may bind to human estrogen receptors (ER-α and ER-β) in competition with 17-β-estradiol. Thus, ZEA, α-zearalenol, and β-zearalenol have estrogenic activity. Previous studies have shown that the estrogenic activity of α-zearalenol is three times that of ZEA, while the estrogenic activity of β-zearalenol is similar to ZEA [7]. ZEA and its derivatives interact with animal estrogen receptors. They are defined as estrogenic mycotoxins. Their toxicokinetics, toxicity, and estrogenic effects have attracted extensive attention [8].

Zearalenone mainly exists in moldy food and crops. It is less sensitive to environmental changes and heat treatment, and thus remains stable during food storage and processing [9]. ZEA mainly contaminates grains, such as corn, wheat, rice, barley, sorghum, soybean, oat, and their products; animal milk may be contaminated when their feedstuff has a high ZEA concentration [10]. Additionally, poultry inevitably contains ZEA in vivo when fed with feedstuff and vegetable oils that contain or contact with mold-contaminated ingredients [2,11,12].

With global climate change and increased environmental pollution, the risk of fungal contamination to food and crops has risen significantly. Therefore, mycotoxin contaminations have become a global issue. ZEA is widely absorbed by rabbits, rats, mice, pigs, poultry, and humans by the oral intake of contaminated grains, foods, and feeds [13]. As ZEA is widely distributed and has high estrogen-like activity, it brings high health risks to either humans or livestock. This review will briefly introduce zearalenone’s toxicokinetics and summarize its toxicity-related studies and current research progress on its endocrine-disrupting effect and intestinal health impacts, thus, providing references for the follow-up research.

## 2. Toxicokinetics of Zearalenone

The toxicokinetics of ZEA relate to its in vivo body-entry rate, absorption, distribution, metabolism, and final excretion. Usually, animals obtain ZEA through contaminated feed. ZEA appears through structural changes during absorption and metabolism, mediated by liver metabolic enzymes and intestinal microflora. These changes lead to the reduction, oxidation, and conjugation of ZEA. The toxicokinetics of ZEA have been studied in various animals through different administration methods [2,11,14,15,16].

### 2.1. Absorption and Distribution

Animal studies have shown that ZEA is rapidly absorbed through the gastrointestinal tract after oral exposure [17]. For example, pigs, rabbits, and humans absorb ZEA quickly; however, their absorption rate is hard to investigate due to the high secretion of bile acids (after being absorbed by the small intestine, part of ZEA will return to the small intestine for secondary absorption through the excretion of bile acids) [13]. About 80–85% of ZEA can be absorbed by pigs orally at 10 mg/kg bodyweight (bw), while the absorption rate decreased among rodents and poultries [4,18].

Several studies have found that ZEA is widely distributed in animal tissues and slowly scavenged [18,19,20]. The liver is the main organ for ZEA deposition [17]. Besides, ZEA distributes to organs or tissues, including the kidney, intestine, adipose tissue, and reproductive organs (uterus, testis, and ovary) [21].

### 2.2. Metabolism and Excretion

The major organs involved in the biotransformation of ZEA are the liver and gut. Target organs of estrogen, such as the ovary, can also convert ZEA, known as the steroid metabolism. This reaction is catalyzed by the intraovarian steroid hydrogenase, 3α- or 3β-hydroxysteroid dehydrogenase (HSD) enzyme. In the liver, the major metabolic pathway of ZEA consists of two steps: the first step is the reduction reaction, whereby ZEA reduces to two non-corresponding stereoisomers, α- or β-zearalenol (zearalenol, ZOL), and then, by the reduction of the C-6′ keto group, ZEA reduces to α- or β-right cyclo-tetra decanol (zearalanol, ZEL). Therefore, it produces at least four metabolites of ZEA: α-ZOL, β-ZOL, α-ZEL, or β-ZEL [19]. These metabolites refer to the reductive metabolites of ZEA in the phase-I metabolism process. Distinctive differences in the phase-I metabolism are found among different animal species. A study on different animal liver microsomes showed that α-ZOL/β-ZOL varies with species. For example, α-ZOL/β-ZOL is high for pigs and humans. However, β-ZOL is predominant in poultry and ruminants [14].

After the phase-I metabolism, the ZEA and its metabolites bind to glucuronic acid in phase-II metabolism. In this step, the toxin and its metabolites are conjugated by uridine diphosphate glucuronyltransferase. Thereby, it forms modified or masked metabolites such as derivatives conjugated with glucose, sulfate, or glucuronide [22].

Part of ZEA and its metabolites, and most of its conjugated compounds produced by phase-II metabolism, are excreted in urine or feces. For example, young female rats were orally dosed with 1 or 10 mg/kg bw ZEA, and about 55% of the toxin was excreted by feces, while another 15–20% was excreted by urine [15]. Likewise, rabbits mainly excreted the toxin through urine, and White Leghorn hens excreted most of the toxins in urine 72 h after the administration of ZEA [17]. A total of 301 urine samples collected from healthy volunteers aged 0–84 years in China were analyzed to determine the total and free ZEA biomarkers, respectively. ZEA, ZOL, α-ZEL, and β-ZEL were detected in 71.4% of the samples at levels of 0.02–3.7 ng/mL after enzyme hydrolysis. Adolescents had higher exposure than children, adults, and the elderly [23].

Bile excretion and entero-hepatic circulation (EHC) are the major ZEA excretion and reabsorption processes in most mammals [18,24]. ZEA-glucuronide derivations are abundantly collected in bile, reabsorbed, and metabolized by intestinal mucosal cells. Again, the toxins enter the portal vein blood, liver, and systemic circulation, where ZOL with high estrogenic activity may be formed. The reabsorption process affects the metabolized and endocrine balance, increasing ZEA’s retention time, prolonging the duration of toxic effects, and delaying its elimination. Studies have shown that ZEA in pig bile reaches the peak 2–6 h after the intravenous injection of ZEA [25]. In addition, due to the extensive glucuronidation of ZEA and its reduced metabolites, it was highly susceptible to EHC, which can have an important impact on the terminal elimination half-lives. This impact was apparent, as demonstrated by a decreased elimination half-life of 2.63–86.6 h [18,25,26]. Figure 2 is a description of phase-I, II metabolism in liver and the hepatointestinal circulation of ZEA.

## 3. Research Progress on the Toxicity of Zearalenone

Zearalenone is one of the top five agriculturally important and of greatest concern mycotoxins [27]. Its toxicity is mainly manifested in the following aspects: reproductive toxicity, hepatotoxicity, immunotoxicity, genotoxicity, and carcinogenicity. Several acute toxicity studies have given oral LD50s of ZEA, which are above 2000, 4000, and 5000 mg/kg bw in mice, rats, and Guinea pigs, respectively [28]. The no-observed-effect level (NOEL) of ZEA in pigs and rats shown by 90-day sub-chronic oral toxicity studies were 40 and 100 μg/kg bw, respectively [2,29]. Since ZEA has an estrogen-like structure, it binds to various estrogen receptors (ERs). Therefore, low-dose ZEA interferes with the physiological–metabolic response and affects the vital functions of the body. Reproductive toxicity is one of ZEA’s main toxic effects, which causes reproductive disorders in various animals. Besides, ZEA is a potential carcinogen. The International Agency for Research on Cancer (IARC) classifies ZEA as the first Class 3 carcinogen [29,30]. Existing studies indicate that ZEA induces genotoxicity by DNA fragmentation, micronucleus formation, DNA adduct formation, chromosomal aberrations, and apoptosis [31,32,33]. Additionally, ZEA induces liver lesions accompanied by cancer development. Moreover, ZEA is immunotoxic and nephrotoxic. It causes changes in immune parameters and chronic progressive nephropathy both in vivo and vitro [34,35,36,37].

Previous studies have broadly focused on the general toxicity of ZEA [34,38]. For example, sows fed with diets containing 1.3 mg/kg ZEA have indicated systemic toxicity in blood biochemistry. ZEA inducted decreased serum platelets, hemoglobin, triglycerides, high-density lipoprotein, and increased liver transaminase activity in vivo and caused histopathological changes in the liver and renal function damages [30]. Exposure to ZEA in mice resulted in blood toxicity by increasing alanine transaminase (ALT), alkaline phosphatase (ALP), and aspartate transaminase (AST), accompanied with decreases in total protein and albumin levels [39]. The long-term toxicity and carcinogenicity of ZEA were determined in Wistar rats by adding the toxin to their daily diet (0.1, 1, and 3 mg/kg). The results indicated a significant decrease in body weight gain among male rats treated with middle-dose and high-dose ZEA. The high-dose-treated group showed an increased liver weight and the middle-dose- and high-dose-treated rats showed a significant uterine weight increase [38]. The above results explored the general toxicity of ZEA among different species, and only characterized the basic phenotypic indicators. On this basis, the toxicological effect of ZEA has been further and systematically discussed.

With more attention paid to the intestinal environment and chronic metabolic diseases in recent years, molecular biology methods were applied to explore ZEA’s impact on intestinal health and endocrine homeostasis. It is found that ZEA may damage intestinal health and interfere with endocrine stability. This state leads to a series of metabolic disorders, and the potential effects of long-term low-dose exposure to zearalenone on the body need special attention. Figure 3 summarizes major toxicity and impact mechanisms of ZEA. The following summarizes the toxicity of ZEA by reviewing the experimental studies in recent decades.

### 3.1. Reproductive Toxicity

ZEA and its derivatives have estrogen-like effects in rats, mice, rabbits, and Guinea pigs. Toxicity to germ cell development and embryonic development in animals or humans is mainly attributed to the following four mechanisms: (1) binding to estrogen receptors as an estrogen compound, causing direct damage to germ cells and organs; (2) destroying the blood–testis barrier and damaging germ cells; (3) increasing oxidative stress and destroying the body’s antioxidant defense system; (4) and promoting cell apoptosis through DNA damage, causing inflammatory responses and leading to a hormone secretion disorder [40]. ZEA induces reproductive dysfunction (infertility or reduced fertility), ovarian and uterine dilation, vaginal prolapse, vulvar swelling, decreased sperm count, serum testosterone, and progesterone levels in rats, mice, pigs, and cattle. Additionally, ZEA may reduce fetal weight when exceeding its critical dose [2,41]. In addition, ZEA causes altered oocyte and follicle development, and preterm birth and miscarriage in mice [42]. Likewise, 42-day-old weaned piglets were treated by ZEA through daily diets to investigate its effects on growth performance, reproductive organs, and immune function. It was found that the piglets’ vulvae length, width, and area were significantly increased; their hypothalamic–pituitary–ovarian axis was disrupted; and the piglets’ estradiol, progesterone, luteinizing hormone, follicle-stimulating hormone levels, and reproductive performance were significantly reduced [43].

### 3.2. Hepatotoxicity

In addition to the reproductive organs, the liver is another target organ for ZEA toxicity. ZEA may change the activity of liver enzymes, the degree of lipid peroxidation, the content of liver protein, the antioxidant capacity, and the inflammatory response, which leads to hepatotoxicity. Additionally, ZEA may cause DNA damage to hepatocytes and severe damage to liver function. The liver damage extents can be assessed by testing specific liver enzymes. A quantity of 40 mg/kg bw ZEA intake increases serum ALP, AST, and ALT levels in mice [34]. After 7 and 14 days of exposure to 100 μg/kg bw ZEA, the serum ALT, ALP, AST, and γ-glutamyltransferase (GGT) activities increased in rabbits, indicating ZEA’s liver toxicity [44]. ZEA increased serum ALT, ALP, and AST activities in rats dose-dependently, and decreased the serum level of total protein and albumin levels [45]. ZEA exposure (40 mg/kg bw) significantly increased the BALB/c mice’s liver tissue malondialdehyde (MAD) levels, protein carbonyl generation, catalase, superoxide dismutase (SOD) activities, expression of the heat-shock protein (HSP 70), and caused oxidative stress in liver tissue [46].

The combination of ZEA and its reduced products (α-ZOL and β-ZOL) has synergistic toxic effects on HepG2 cells at high concentrations, which can significantly change the expression of inflammation-related genes IL-1β, TNF-α, and IL-8. Additionally, the toxin reduces cell viability and triggers inflammatory responses [47]. Besides, ZEA regulates cytochromes (HCYP) in the liver. Existing research treated three types of hepatocytes with ZEA in different concentrations to examine the gene expression of the cytochrome P450 family. The study found that ZEA activated the mRNA levels of human PXR, CAR, and AhR. At a concentration as low as 0.1 μM, the target genes involved in the metabolism of its phase-I reaction were mainly CYP3A4, CYP2B6, and CYP1A1. The transcriptional expression of CYP3A, CYP2B, and CYP1A were regulated by the nuclear receptor PXR and (or) the CAR pathway and AhR-mediated pathway, respectively [48]. Another study confirmed these findings and explored the effect of metabolism-related molecules in the liver phase-II reaction. The authors found that 20 μg/mL ZEA significantly activated Nrf 2 and increased the mRNA level of UGT1A. However, 40 μg/mL ZEA did not increase the mRNA levels of Nrf 2 and UGT1A. It suggested that ER stress caused by high ZEA concentrations may be one of the reasons for the decreased expression of phase-II reaction metabolism-related molecules (Nrf 2 and UGT1A), disrupting the normal hepatocyte detoxification process and reducing the expression of phase-I/II reaction enzymes [49].

Recent studies have explored ZEA’s effect on the overall methylation, histone modification, and transcriptional profiles of chromatin-modifying enzymes in HepG2 cells from the level of cellular epigenetics and gene expression. It was found that ZEA and α-ZOL at concentrations of 1, 10, and 50 μM significantly increased the overall methylation and histone modification levels (H3K27me3, H3K9me3, and H3K9ac) and changed the AhR, LXRα, PPARα, PPARγ, L-Fabp, LDLR, Glut 2, Akt 1, and HK2 gene expressions associated with nuclear receptors and metabolic pathways. PPARγ, which is related to lipid metabolism, was the critical regulator. Further research found that the gene’s promoter methylation of PPARγ reduced significantly, indicating that epigenetic modification may be one of the ways that ZEA impacts the metabolic pathways [50].

### 3.3. Immunotoxicity

ZEA impairs immune function, causes monocyte proliferation, induces immune organ damage, and modulates the levels of inflammatory factors and immunoglobulins. Leukocytes, NK cells, proinflammatory cytokines, immunoglobulins (IgG and IgM), some subtypes of B and T cell (CD3, CD4, and CD8) levels decreased in BALB/c mice treated with 40 mg/kg bw ZEA for two weeks [51]. Additionally, ZEA resulted in a decreasing expression of tumor necrosis factor-alpha (TNF-α) in mice and pigs. Treating ovariectomized rats with 3.0 mg/kg ZEA for 28 days damaged their lymphoid organs, modulated immune responses, caused thymus atrophy, and inhibited T cell-mediated immune responses [52]. In primary splenic T lymphocytes of mice activated with concanavalin, ZEA treatment inhibited T lymphocyte activity, caused intracellular and surface ultrastructural damage, and inhibited the expression of CD25 and CD278. Additionally, ZEA inhibited the synthesis of some effector molecules and over-activated the MAPK signaling pathway to promote the apoptosis of T lymphocytes and thus inhibited immune functions [53]. ZEA has a cytotoxic effect on human leukemic cell lines, HL60 (promyelocytic) and U937 (monocytic), and peripheral blood mononuclear cells (PBMCs). ZEA causes hypodiploid peaks and G1-phase arrest. Its mechanism of inducing apoptosis is to decline the mitochondrial membrane potential, activate caspase-3 and -8, generate reactive oxygen species, cause endoplasmic reticulum stress, and release cytochrome c from mitochondria into the cytoplasm [35]. Additionally, ZEA inhibits the chemotaxis of T cells by inhibiting the expression of cell adhesion and migration-related proteins. An existing study activated the spleen lymphoid T cells of BALB/c mice with concanavalin and administered the cell with ZEA (10, 20, and 40 μM). The results showed that ZEA treatment caused ultrastructural damage on the surface and interior of T cells, inhibited T cell chemotaxis mediated by CCL 19 or CCL 21, disrupted the balance of T cell subtypes, and inhibited T cell chemokines. The decreased expression of receptors inhibited the secretion of chemokines such as RANTES and MIP-1α from T cells, inhibited the migration of the cells, and limited their immune functions [54].

### 3.4. Genotoxicity

ZEA has been reported to be genotoxic. It may lead to an increased production of reactive oxygen species, chromosome aberration, lipid peroxidation, DNA adduct and breakage, SOS repair, cell apoptosis, and so on. ZEA treatment in cultured bovine lymphocytes resulted in chromosomal aberrations (CAs) and sister chromatid exchanges (SCE), as well as the reduced mitotic index (MI) and cell viability, mediating programmed cell death [55]. Another study in BALB/c mice with a single injection of ZEA at a dose of 2 mg/kg bw detected multiple DNA adducts in the mouse kidneys [56]. ZEA binds to 17-β-estradiol receptors, induces lipid peroxidation in cells, causes cell death, and inhibits protein and DNA synthesis. ZEA treatment was found to significantly reduce CaCo-2 cells’ viability, inhibit protein and DNA synthesis, and cause a significant increase in the lipid Peroxidation MDA [57]. A quantity of 40 μM ZEA was found to inhibit the proliferation of porcine intestinal epithelial cells IPEC-J2, arrest the cell cycle at the G2/M phase, and affect the cellular transcriptional profile. A total of 783 differentially expressed genes (DEGs) were identified. The KEGG pathway analysis revealed that PERK regulates gene expression. Toll-like receptors stimulate signaling pathway, mitosis, metaphase and anaphase, DNA replication, and the G2/M checkpoint were involved in cell cycle pathways. Eleven critical genes related to the G2/M checkpoint were verified by the qPCR method, indicating that the G2/M checkpoint is the most important signaling pathway affecting the cell cycle [58]. Additionally, ZEA at this concentration (40 μM) was reported to induce apoptosis in the kidney cells that originated from male Swiss mice. The mRNA expression and protein levels of Bax, caspase-12, caspase-3, Bip (possible targets), CHOP, and JNK increased, while the mRNA expression and protein levels of Bcl-2 decreased [59]. Likewise, ZEA treatment to BALB/c mice increased their renal cell caspase-3 activity and relevant mRNA levels (MDA, IL-10, IL-6, TNF-α, and Bax). Additionally, ZEA treatment decreased the mice’s total antioxidant activity (TAC) and relevant mRNA expressions (GSH-Px, CAT, and Bcl-2) indicating that ZEA induced oxidative stress in renal cells and caused genotoxic effects [60]. Besides, ZEA was found to induce ROS-mediated cell cycle arrest and apoptosis in mouse Sertoli cells via endoplasmic reticulum (ER) stress and the ATP/AMPK pathway. Treating TM4 cells with 0–30 μM ZEA led to ROS accumulation, which induced ER stress, inhibited cell proliferation through the ATP/AMPK pathway, affected cell cycle distribution, and induced apoptosis [61]. Additionally, ZEA caused autophagy, apoptosis, and destruction of the cytoskeleton structure in mouse TM4 cells, through oxidative stress, ER stress, PI3K-AKT-mTOR, and MAPK signaling pathways [62]. In rat insulinoma INS-1 cells, 50, 100, and 500 μM ZEA were found to induce NF-κB p65 activation, which promotes the activation of NLRP 3 inflammatory bodies in INS-1 cells, formats inflammatory response and phagocytosis, and induces NLRP 3-dependent inflammatory cell death [49].

### 3.5. Carcinogenicity

In addition to the above toxicity studies, ZEA may induce liver cancer, breast cysts, chronic progressive nephropathy, retinopathy, and cataracts, as it promotes cell proliferation. ZEA may induce liver cancer from liver injuries in mice, which was first reported in 1982. Additionally, it was found that ZEA treatment causes prostate inflammation, mammary cysts, and hepatocyte vacuoles in rats [54]. ZEA resulted in a higher incidence of increased trabecular formation in the femoral bone marrow of Wistar rats at 3 mg/kg bw [38]. Additionally, ZEA was found to stimulate the growth of human breast cancer cells, as the cells have estrogen receptors, which leads to an increased incidence of breast cancer [63]. In 2018, a study examined the effect of 10 and 30 μM ZEA on tumorigenic gene expressions on ovarian granulosa cells obtained from CD1 mice. The results showed that 30 μM ZEA exploration increased the expression of multiple cancer-associated genes, such as the Hippo signaling pathway and related genes (CCND 1, SMAD 3, Tead3, YAP 1, and WWTR1). Furthermore, immunohistochemistry explored that 30 μM ZEA treatment increased the protein levels of YAP 1, WWTR1, and CCND 1, resulting in abnormal cell morphology and an increased tumorigenic risk [64].

### 3.6. Gastrointestinal Health

In addition to basic toxicological research, mycotoxins and gut health have received increasing attention in recent years. When humans and animals ingest mycotoxins, the gut is the first to be affected. Mainly, the toxin may affect gut histomorphology and gut microbes. The intestinal barrier refers to the physical, chemical, biological, and immune barriers. ZEA may destroy these barriers, thus resulting in decreased intestinal resistance to toxins and affecting the immune function. ZEA’s effect on intestinal morphology and histopathology has been accessed in 21-day-old SD rats. After a 4-week treatment with ZEA at different concentrations (0.2, 1.0, and 5.0 mg/kg bw), significant intestinal villi and glands damages, and mucosal epithelium and lamina propria separations were observed. The microvilli coefficient and length/recess depth of jejunal villus decreased, and the intestinal permeability increased significantly. The remarkable expression increases of IL-1β, IL-6, TNF-α, IFN-γ, and CCL 20 cytokines, and the decrease of IL-10, indicated that ZEA caused intestinal inflammation and reduced the expression of tight junction-associated proteins. Although ZEA treatment slightly increased the α-diversity of gut microbes in terms of cecal microflora diversity composition, it significantly decreased the β-diversity, indicating that the integrity of the intestinal barrier and the balance of intestinal flora was disrupted [65]. It was found that the oral admission of 4.5 mg/kg bw ZEA for a week increased the mRNA expression of inflammasome NLRP 3, pro-interleukin-1β (pro-IL-1β), and pre-IL-18 (pro-IL-18) from 0.5-fold to 1-fold in BALB/c mice. Additionally, their IL-1β and IL-18 release increased by one-fold. Loose stools were observed clinically. Besides, the histology showed marked inflammatory cell infiltration and colon tissue damage. Additionally, this study summarized the potential mechanism of intestinal inflammation caused by ZEA: (1) ZEA initially had a direct toxic effect on the epithelial barrier, enhancing the accumulation of ROS, and then, macrophages were activated by ROS, thereby enhancing the transcription of pro-IL-1β and pro-IL-18; (2) the induction of caspase-1 activation by the NLRP 3 inflammasome cleaves pro-IL-1β and pro-IL-18 into biologically active forms that initiate the intestinal inflammatory cascade [66]. In IPEC-J2 cells, ZEA increased lactate dehydrogenase (LDH) activity, cell permeability, and reactive oxygen species levels. It reduced the expression of intestinal immune barrier-associated immunoglobulin (IG A\G\M) and intestinal physical barrier-related genes (PBD-1, PBD-2, MUC-2, ZO-1, occludin, and claudin-3). Thus, ZEA induced oxidative stress and intestinal epithelial barrier damage in IPEC-J2 cells [67].

### 3.7. Endocrine-Disrupting Effects

The estrogenic effects of ZEA have been extensively demonstrated in many species. ZEA binds to estrogen receptors (ER-alpha and ER-beta) with a high affinity to ER-alpha and activates the transcription of estrogen-responsive genes. Therefore, it acts as an endocrine disruptor through ERs in vivo. The endocrine disturbance caused by ZEA is closely related to the release of pituitary hormones. Zearalenone and its metabolites have been reported to regulate luteinizing hormone (LH) production by modulating the estradiol receptor GPR 30, and dietary exposure to prepubertal rats decreased the premature activation of Kisspeptin–Gpr54–GnRH hypothalamic signaling [68]. Oral administration of 20 mg/kg bw ZEA to male 10-week-old rats for five consecutive weeks resulted in a significant increase in serum prolactin concentration, whereas their serum luteinizing hormone, follicle-stimulating hormone levels, and germ cell numbers remained unchanged [69]. ZEA was found to promote follicle growth through an ERs/GSK-dependent Wnt-1/β-catenin pathway in pigs [70]. Additionally, ZEA was found to inhibit the testosterone synthesis in mouse Leydig cells by inhibiting the orphan nuclear receptor Nur 77 [71]. Besides, ZEA induces precocious puberty in female rats through early stimulation of the hypothalamic KiSS1/GPR54 pathway [72]. In addition to the toxin’s estrogen-like structure-leaded chronic toxicity and sub-chronic toxicity on the endocrine disorder, which causes reproductive disorders, the mentioned studies explored the effect of ZEA on body metabolism from the perspective of endocrine disruption. However, the intuitive ability of endocrine disruption on obesity, abnormal blood lipids, blood sugar, and blood pressure remains unknown. Few studies have been performed to verify the relationship between metabolic disease and ZEA toxicity in terms of metabolism interference. Some researchers found that obesity increased the ovarian responses to ZEA (40 μg/kg bw) among seven-week-old female wild-type nonagouti KK.Cg-a/a mice (lean) and agouti lethal yellow KK.Cg-Ay/J mice (obese) [73,74].

The molecular mechanisms and related pathways involved in these studies may be the key factors to the association of zearalenone with metabolic diseases. However, few studies have proved their relationships directly, and it is difficult to carry out epidemiological investigations and cohort studies when studying their correlations with metabolic diseases. Therefore, at present, those studies strongly rely on animal experiments. Table 1 summarizes some pathways involved in different types of toxicity of Zea.

## 4. ZEA Exposure and Risk Assessment

### 4.1. ZEA Generations

ZEA and its derived toxins are mainly ingested by the oral admission of contaminated crops [2]. Therefore, the exposure risk to the toxins is closely related to the food contamination levels and the proportion of grains in the diet. In addition, temperature, precipitation, and carbon dioxide concentrations are critical to the mycotoxin’s formation in crops.

#### 4.1.1. Temperature

The mild temperatures during grain harvesting aid the growth of *Fusarium* species and promote toxin production [82]. The fungi performed high growth and toxin production abilities at both 28 °C and 15 °C [83]. Likewise, another study found that the optimum temperature for *F. graminearum* is from 15 to 25 °C [84]. However, these fungi do not tolerate high temperatures. Multiple studies have shown that ZEA was not produced over 37 °C [83,85]. Additionally, the *Fusarium*’s perithecia were only found to mature at 20–25 °C [86]. Therefore, although some F.graminearum strings native to the tropics grow slowly at high temperatures of 40 °C, ZEA contamination on crops is more common in areas with cooler climates, such as Europe [87].

#### 4.1.2. Humidity (Water Activity)

High water activity enables the growth and toxigenicity of fungi. For example, F. incarnatum grows in sorghum and produces ZEA with a water activity range of 0.91–0.99; *F. graminearum* PH1 and *F. graminearum* F1 produce ZEA in maize flour and maize kernel with a water activity greater than 0.925 [88]. High precipitation during crop flowering promotes *Fusarium* infestation of crops, while continuous pre-harvest precipitation means that the optimum humidity for fungal growth and mycotoxin production within the grain is prolonged [82,89]. For example, a 10-year global survey found significantly higher ZEA concentrations in maize harvested in Central and Southern Europe in 2014, corresponding to persistently high precipitation during the maize silking and pre-harvest periods; in contrast, low precipitation before harvesting in China’s corn core-planting areas in 2013 resulted in low average ZEA levels in East Asia [82]. Additionally, crops were delayed during a wet harvest in the UK in 2008, resulting in 29% of wheat exceeding the European limit of 100 µg/kg ZEA for unprocessed grains [90]. The World Health Organization (WHO) reported that climate change might cause grains’ harvested water activities to be 12–14% higher than suitable for storage [91].

#### 4.1.3. CO_2_

Increased CO_2_ in the environment enhances the pathogenicity of *Fusarium* and the susceptibility of crops to pathogens. It was found that wheat grown at 780 ppmv CO_2_ was more susceptible to *Fusarium* head blight (FHB) and Septoria tritici blotch (STB) caused by Zymoseptoria tritici and *Fusarium graminearum* than wheat grown at 390 ppmv CO_2_ [92]. Similarly, another study confirmed that *F. graminearum* is well tolerant to CO_2_ up to 1200 ppm [93].

#### 4.1.4. Climate Change

Currently, global climate change is causing elevated CO_2_ levels, unconventional rainfall, warmer temperatures, and extreme weather [94]. Elevated CO_2_ levels and high precipitation during grain flowering and harvest are predicted to increase the risk of ZEA contaminations. Additionally, global climate change may lead to an increase in agricultural pests [95]. Insect nibbling destroys the plant’s protective barrier and promotes the colonization and growth of fungi [96]. Additionally, increased insect entrainment and air movement facilitate the spread of toxigenic fungal spores between plants [97]. A modeling prediction in South Korea shows that the risk of rice ZEA pollution will gradually increase in the following decades due to climate change.

#### 4.1.5. Other Factors

Other factors affect ZEA levels in crops. On the one hand, pre-harvest (e.g., farming rotations and weeding) and post-harvest (e.g., dehydration, fermentation, and hulling) practices affect the mycotoxin content in grains [90,98]. On the other hand, the crops’ susceptibility to the *Fusarium* species differs. For example, the ZEA content in amaranth grains (11.1 μg/g) was significantly lower than that of maize (600–2585 μg/g) and wheat (2485 μg/g) under similar culture conditions [99]. In addition to the high risk of ZEA in grains, wastewater discharges allow ZEA to migrate from fields to water [100,101]. Several studies have reported detecting ZEA in treated industrial wastewater, surface water, drinking water, and feed water [102,103,104]. Exposure to the ZEA-polluted environment affects aquatic animal health, disrupts the ecological balance, and directly or indirectly endangers human health [105]. However, a comprehensive survey of ZEA contamination and human exposure in global aquatic environments is still lacking.

### 4.2. Exposure and Allowable Limits

A risk assessment of human ZEA exposure is important as it is relatively common. Existing studies calculated the estimated daily intake (EDI) of ZEA in Europe and America based on local ZEA contamination levels and daily grain consumption. The result showed that the EDI for adults ranged between 0.8 and 25 ng/kg bw, and between 6 and 55 ng/kg bw for children [98,104]. The estimated ZEA exposure may be lower than the actual intake due to the fact that most detection methods cannot detect the modified and masked ZEA. The modified form of ZEA was reported to be 1.5-fold higher than the unmodified ZEA in barley and 0.5-fold higher in oat and wheat [98]. However, population samples reflect the exposure level of ZEA more truthfully. Researchers analyzed the biomarker concentrations, urine output, body weight, and excretion rates of 301 urine samples of healthy volunteers aged 0–84 in China. The result indicated that the possible daily intake (PDI) of ZEA in China was 41.6 ng/kg bw; ZEA exposure of adolescents was higher than that of children, adults, and the elderly [90].

ZEA presents in crops and processed food in a variety of regions globally, such as Europe, Asia, and Africa [88,99,100,106,107]. Countries with a mild and humid climate have more serious ZEA pollution in crops. A total of 16 countries, including South Korea, Thailand, China, Japan, and other Asian countries, have set limits on ZEA. The EU limits ZEA in ready-to-eat corn, corn snacks, breakfast cereals, and processed cereals for infants and young children to 100 μg/kg; China limits ZEA in cereal and cereal products (such as wheat, wheat flour, corn, and cornmeal) to 60 μg/kg in GB 2761–2017. Additionally, various international organizations have given recommendations on the ZEA dietary exposure. The Food and Agriculture Organization of the United Nations (FAO) and the Joint Expert Committee on Food Additives (JECFA) under the World Health Organization (WHO) set the provisional maximum tolerable daily intake (PMTDI) of ZEA to 0.5 μg/kg bw [62]. Likewise, the European Food Safety Authority (EFSA) sets tolerable daily intake (TDI) to 0.25 μg/kg bw [83].

### 4.3. Disintoxication and Detoxification

Although traditional processing procedures such as shelling, soaking, milling, fermentation, heat treatment, or extrusion contribute to eliminating ZEA, they cannot remove the mycotoxins from food and feed thoroughly [108]. Detoxification of ZEA mainly relies on physical, chemical, and biological approaches.

From the physical point of view, substances such as activated carbon, montmorillonite, nanomaterials, and graphene oxide exhibit good ZEA adsorption and immobilization properties [109,110]. A study to optimize montmorillonite with a mixture of binary surfactants (polyoxymethylene ether and lauroamidopropyl betaine) found that the modified montmorillonite was more hydrophobic and more effective in adsorbing ZEA [111]. Additionally, ZEA is highly polar and can be extracted with 90% acetone, 80% isopropanol, methanol, and acetonitrile [112,113]. Solvent extraction has been broadly applied in eliminating mycotoxins from feed [114]. Additionally, heating, irradiation, and cold plasma have ZEA degradation potential [109]. Cold plasma has the best degrading ability to mycotoxins (100%), while the detoxification efficiency of heating is lower (36–41% in corn) [115,116,117].

Chemical detoxification destroyed the chemical structure of ZEA. For example, ozone effectively reduced ZEA contaminations in cornmeal (up to 62.3%); however, its oxidizing properties altered the peroxide value, fatty acid profile, and gelatinization properties [118]. Thus, it harms the nutritional value and flavor of grains.

The biological approach utilizes microorganisms to adsorb or enzymatically detoxify ZEA. Some yeast and *Lactobacillus* strains are cytologically compatible with ZEA, thus adsorbing ZEA and reducing its bioavailability [109,119,120]. Additionally, microorganisms such as *Bacillus* subtilis were found to change the molecular structure of ZEA during their metabolic processes so that the estrogenic effect of ZEA can be reduced [121,122].

Currently, chemical detoxification methods are only used in industrial production. The physical adsorption and biological detoxification approaches are applicable both in vivo and in vitro [123,124,125]. However, although some physical adsorbents, such as aluminosilicates, exhibited detoxification effects in vitro, they were less effective in vivo, thus indicating the differences in detoxification potentials in vivo and in vitro [126]. Consequently, animal experiments are critical for the in vivo detoxification ability assessments.

Detoxification in vivo aims to alleviate or antagonize the toxic effects of ZEA from the molecular mechanism and pathway regulation, using natural products, probiotics, and other means. For example, dietary supplementation with adenosine 5‘-monophosphate (AMP)-activated protein kinase (AMPK), hydroxy-delta-5-steroid dehydrogenase, 3 beta- and steroid delta-isomerase 1 (HSD3B1), and estrogen receptor 1 (ESR1) loci, ameliorated ZEA-induced hepatotoxicity and reproductive toxicity in rats [127]. N-acetylcysteine alleviated its toxic effects on porcine Sertoli cells (SCs) by reducing ZEA-induced oxidative damage and apoptosis [128]. The combination of four probiotics, *Bacillus subtilis*, *Lactobacillus casein*, *Candida utilis*, and *Aspergillus oryzae* antagonized the toxicity of ZEA by alleviating intestinal inflammation and apoptosis [129]. Various natural products such as allicin, chitosan, melatonin, breviscapine, and resveratrol have been reported to have ZEA detoxify potentials [130]. Further studies are required on the in vivo detoxification of ZEA by natural products and probiotics.

## 5. Conclusions

ZEA and its metabolites are potential harms to the health of animals and humans through the food chain and are increasingly recognized by people. Long-term high-dose exposure to ZEA may cause severe toxic effects in mammals, disturb the reproductive system, and induce endocrine disorders. Thus, it may cause economic losses in the breeding industry. Hence, the development and application of the detoxification approach are required, based on several existing toxicological studies. However, long-term low-dose exposure to ZEA causes endocrine disorders, which may lead to metabolic disorders, and increase the risk of metabolic syndrome-related diseases. Despite this, ZEA’s estrogen-like impacts on in vivo metabolic reactions require further studies. Studies in recent decades have mostly been conducted in animals with high-dose short-term exposure, ignoring the subtle chronic effects of low-dose long-term exposure to toxins that are closer to human exposure. The susceptibility and adverse effects of ZEA for special populations, such as infants, pregnant women, hypertensives, diabetics, the obese, and postmenopausal women, should be elucidated as soon as possible, so as to further guide the formulation and development of toxin residue limits in related foods. In future research, special attention should be paid to the effects of toxin levels close to human exposure, and in the selection of animal models, and the use of pathogenic means that are significantly different from humans (such as feeding with extremely high-fat-content diets) should be avoided as much as possible so as to avoid drawing data and conclusions that do not apply to humans.

## Figures and Tables

**Figure 1 toxins-14-00386-f001:**
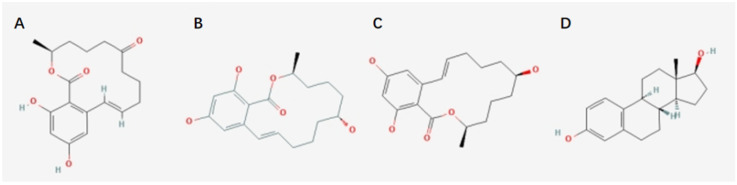
Chemical structure formulas. (**A**) ZEA, (**B**) α-ZOL, (**C**) β-ZOL, (**D**) 17-β-estradiol.

**Figure 2 toxins-14-00386-f002:**
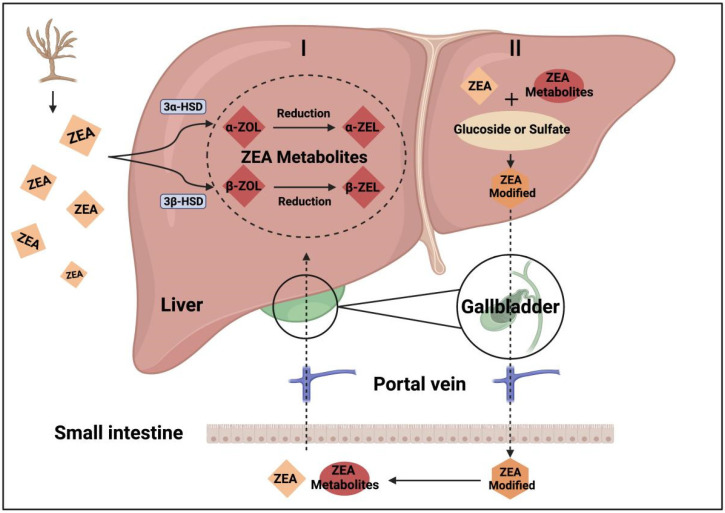
The metabolism and reabsorption of ZEA (created with Biorender.com accessed on 21 March 2022).

**Figure 3 toxins-14-00386-f003:**
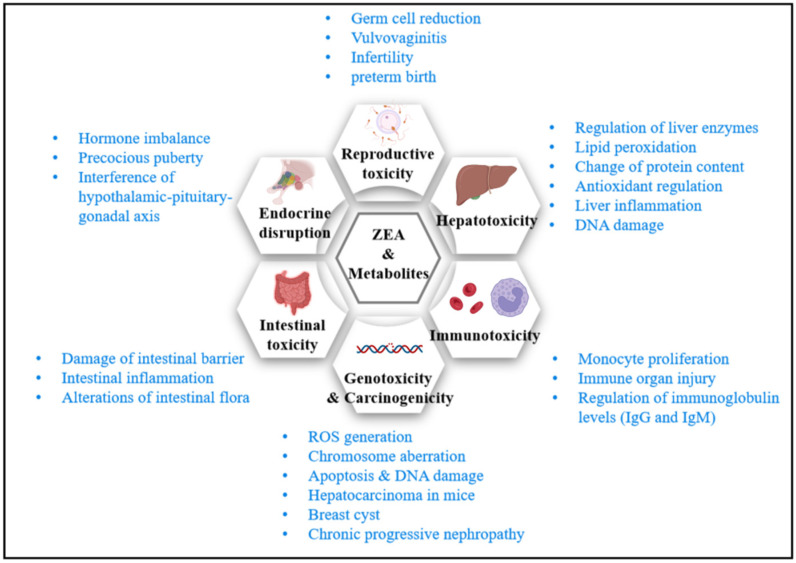
Toxicity of ZEA (created with Biorender.com accessed on 21 March 2022).

**Table 1 toxins-14-00386-t001:** Summary of ZEA-induced toxicity.

Toxic Type	Animal/Cell Model	Dose	Exposure Time	Phenotypic Modulation	Pathway	Reference
Reproductive toxicity	Postweaning piglets	3 mg/kg bw	28 d	Vulvar malformation, decreased immune response, and disorder of the level of serum hormones.	Hypothalamic–pituitary–ovarian axis pathway.	[43]
	TM4 cells	0~30 μM	24 h	TM4 cell autophagy, oxidative stress, and cytoskeletal structure destruction.	PI3K/Akt/mTOR and MAPK signaling pathway.	[62]
	TM3 cells	50 μM	24 h	Decreased cell viability and testosterone concentration, increased LDH, and cell apoptosis.	PI3K/Akt, PTEN/Nrf 2/Bip, and ER-stress signaling pathway.	[75]
	Sertoli cells (SCs)	0~80 μM	24 h	Cell cycle arrest, inhibited SCs proliferation, and cell morphological autophagy.	PI3K/Akt/mTOR signaling pathway.	[76]
Hepatotoxicity	Balb/c mice	40 mg/kg bw	24 h	Increased MDA level, protein carbonylation, SOD activity, CAT activity, and the expression level of HSP70.	Oxidative stress pathway.	[46]
	HepG2 cells	0~100 μM	72 h	Decreased cell viability and the expression of liver inflammation-related factors.	Inhibit inflammatory response and liver immunity.	[47]
	HepG2 cells	0~40 μM	24 h	Decreased cell viability, increased production of ROS, and regulated phase-I/II metabolism, resulting in autophagy and apoptosis.	Oxidative stress, ER-stress, and PERK/eIF2α pathway.	[49]
Immunotoxicity	T lymphocytes	0~40 μM	24 h	Decreased cell viability, damaged cell surface and intracellular ultrastructure of T lymphocytes, and decreased secretion of cytokines, resulting in cell apoptosis.	MAPK signaling pathway, TNF-α-independent JNK signaling pathway.	[53]
	HL-60, U937, PBMCs	0~50 μM	24 h	Decreased cell viability, increased production of ROS, and cell apoptosis.	The death receptor pathway with direct involvement of caspase-8, the mitochondrial pathway, and ER-stress pathway.	[35]
	T lymphocytes	0~40 μM	48 h	Surface and intracellular ultrastructural damage of T lymphocytes.	Chemokines MIP-1α and RANTES secreted by T lymphocytes and chemokine receptor CCR2 and CCR7.	[54]
	T lymphocytes	0~40 μM	24 h	Decreased cell viability and the expression of different activation signals in T cells inhibited the secretion of cytokines.	Co-stimulatory signal and PI3K/Akt/mTOR signaling pathway.	[77,78]
Genotoxicity	Kunming mice	40 mg/kg bw	28 d	Damaged kidney resulting in oxidative stress and renal cell apoptosis.	Bip, CHOP, caspase-12, and JNK signaling pathway	[59]
	TM4 cells	0~100 μM	24 h	Inhibited cell proliferation, cell cycle arrest, and cell apoptosis.	ROS and ER stress, ATP/AMPK pathway.	[61]
Carcinogenicity	INS-1 cells	0~800 μM	24 h	NLRP3 inflammasome activation, decreased cell viability, cell autophagy, and pyroptosis.	NF-κB p65 activation and nuclear translocation.	[79]
	Mouse granulosa cells	10&30 μM	72 h	Changed cell morphology, cell cycle arrest, and increased expression of genes related to tumorigenesis.	Hippo signaling pathway.	[64]
	TM3 cells	0~90 μM	24 h	Decreased cell viability.	Ras, Rap1, PI3K/AKT, Foxo, and AMPK signaling pathway.	[80]
Gastrointestinal health	IPEC-J2 cells	0~80 μM	24 h	Decreased the cell viability and increased LDH activity, and inhibited cell proliferation, resulting in cell cycle arrest.	Pathways involved in the cell cycle G2 phase.	[58]
	SD mice	0~5.0 mg/kg bw	28 d	Impaired intestinal barrier, increased permeability and imbalance of intestinal microbiota, and increased systematic intestinal inflammation.	RhoA/ROCK signal pathway.	[65]
	Balb/c mice	4.5 mg/kg bw	7 d	NLRP3 inflammasome activation and intestine inflammatory.	NLRP3 signaling pathway.	[66]
Endocrine-disrupting effects	Postweaning piglets	0~3.2 mg/kg bw	18 d	Inhibited LH secretion.	Kisspeptin–Gpr54–GnRH Pathway.	[68]
	Postweaning piglets	0~1.5 mg/kg bw	35 d	Inhibited follicles maturation and ovarian development.	ERs/GSK-3β-dependent Wnt-1/β-catenin signaling pathway.	[70]
	SD mice	0~5 mg/kg bw	5 d	Released gonadotropin early—resulted from the advancement of vaginal opening and enlargement of the uterus at the periphery.	Hypothalamic kisspeptin–GPR54 signaling pathway.	[81]

## Data Availability

Not applicable.

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
