# Peer review of "Research Progress of Safety of Zearalenone: A Review"

_toxins, 2022, doi:10.3390/toxins14060386_

Round 1

Reviewer 1 Report

Good job! Since ZEA belongs to important group of mycotoxins in human health and life protection and additionally is of prime concern in significant economical loses in agriculture and animal production. I believe that it is valuable manuscript in knowledge systematize in the metabolite harmfulness.

With full attention I have become acquainted with way of the problem introduction, elucidation and with special emphasis to methodology. This way prepared manuscript will be important knowledge summary and help especially for the beginners in the topic.

The Authors of the manuscript refer to well selected set of publications – very representative references (mostly updated but also referring to older, but very important publications). This is adequate to presented topics.

The manuscript is prepared in good English, is easy in reading and needs only small (cosmetic) corrections and adjustations.

I recommend this manuscript to be published.

Reviewer 2 Report

Dear authors,

This work appears to be well organised, but is not adequately written. There are some points that need to be addressed in order to be accepted. However, in my opinion, in the current form, the manuscript cannot be accepted.

1: I strongly encourage the authors to check the grammar and spelling of the article. If a native speaker could proofread it, it might be helpful. Sometimes, the text lacks any flow or connection.

2: Many paragraphs lack citations.

3: There are several grammatical errors.

4: You should comprehensively describe the sections or works quoted. In most cases, you only described certain results obetained by other authors.

Introduction:

-Honestly, the introduction needs to be rewritten to make the text more free-flowing and tell us a narrative.  You have mixed information on toxicity, chemical characteristic and occurrence in the same paragraph. It is confusing and unclear.

-Further comments are attached.

Reviewer 3 Report

The importance of the presented review is undeniable. Mycotoxins represent a threat to humans and animals and should be well studied. The case of ZEA are even more important due to its hormonal effects. However, the manuscript needs major revision in terms of sentence construction and verb tenses & punctuation. Many examples can be given along the text of confusing sentences or missing the right verb tenses. For example, line 87 “Thereby, form modified or masked metabolites such as derivatives conjugated with glucose, sulfate, or glucuronide.” This sentence misses the verb.

The abbreviation ZOL should be presented in the first reference to Zearalenol (alfa and beta) in the introduction. It appears first in figure 1 as α-ZOL and β-ZOL.

There is no table 1, first table is 2.

Another important topic that could be added to this review is some information about the detoxification options or preventive measures that can be promoted to avoid contamination of food products and undesirable contact with humans and animals.

Round 2

Reviewer 2 Report

I read the article and I believe it is now suitable for publishing.

Reviewer 3 Report

The manuscript has been improved based on the comments previously sent to the authors.